# Impact of Allogeneic Stem Cell Transplant on Safety and Outcomes of Chimeric Antigen Receptor T Cell (CAR-T) Therapy in Patients with Multiple Myeloma (MM)

**DOI:** 10.3390/jcm13206207

**Published:** 2024-10-18

**Authors:** Ayrton Bangolo, Behzad Amoozgar, Lili Zhang, Vignesh K. Nagesh, Imranjot Sekhon, Simcha Weissman, David Vesole, Pooja Phull, Michele Donato, Noa Biran, David Siegel, Harsh Parmar

**Affiliations:** 1Department of Hematology and Oncology, John Theurer Cancer Center, Hackensack University Medical Center, Hackensack, NJ 07601, USA; 2Department of Internal Medicine, Palisades Medical Center, North Bergen, NJ 07047, USA; 3Division of Myeloma, John Theurer Cancer Center, Hackensack University Medical Center, Hackensack, NJ 07601, USA; 4Division of Bone Marrow Transplant and Cellular Therapy, John Theurer Cancer Center, Hackensack University Medical Center, Hackensack, NJ 07601, USA

**Keywords:** Cellular Therapy, CART-Therapy, stem cell transplant, GVHD

## Abstract

**Background:** Allogeneic stem cell transplantation (allo-SCT) has seen limited use in treating multiple myeloma (MM), despite its potential to offer long-term survival or even cure through the graft-versus-myeloma effect. Its limited application is largely due to concerns over serious complications like infections and graft-versus-host disease (GVHD). The possibility of GVHD exacerbation when CAR-T cells are administered to patients previously treated with allo-SCT remains a topic of concern. Ciltacabtagene autoleucel (Cilta-cel) and idecabtagene vicleucel (Ide-cel) are CAR-T therapies that have been FDA-approved for relapsed/refractory (R/R) MM. A recent study using data from the CARTITUDE-1 trial has shown promising safety and efficacy of Cilta-Cel in patients with a prior history of allo-SCT. This report outlines our real-world experience with CAR-T treatment in such patients. The objective of this study is to assess the safety and effectiveness of CAR-T therapy in R/R MM patients who have previously undergone allo-SCT. **Methods:** We conducted a retrospective analysis of adult patients (18–70 years old) with R/R MM treated with CAR-T therapy as part of an institutional IRB-approved protocol. Data were collected on safety and efficacy outcomes from the institution’s records. Adverse events (AEs) were evaluated using the National Cancer Institute Common Terminology Criteria for Adverse Events (NCI-CTCAE) version 5.0. Cytokine release syndrome (CRS) and immune effector cell-associated neurotoxicity syndrome (ICANS) were graded based on American Society for Transplantation and Cellular Therapy (ASTCT) criteria. Efficacy metrics included overall response rate (ORR) and progression-free survival (PFS), analyzed through the Kaplan–Meier method, with PFS defined as the time from CAR-T initiation to disease progression or death. **Results:** Of the 56 patients treated with CAR-T therapy, 8 (14.3%) had previously undergone allo-SCT. These patients had a median of seven prior therapy lines (LOTs), compared to five LOTs in the non-allo-SCT group (*p* = 0.04). CAR-T infusion occurred a median of 98.8 months after allo-SCT, with a range from 57.9 months to 178.5 months. CRS occurred in 87.5% of the allo-SCT group versus 77.1% in the non-allo-SCT group (*p* = 0.48). One patient in the allo-SCT group developed hemophagocytic lymphohistiocytosis (HLH), requiring anakinra. At a median follow-up of 4.8 months, the ORR was 87.5% in the allo-SCT group versus 75% in the non-allo-SCT group (*p* = 0.4). Median PFS had not been reached for the allo-SCT group at the time of analysis compared to 11.9 months in the non-allo-SCT group (*p* = 0.5). No treatment-related mortality or acute GVHD was noted in the allo-SCT cohort. **Conclusions:** The study suggests that prior allo-SCT does not adversely affect the safety or efficacy of CAR-T therapy in patients with R/R MM. These findings highlight the need for further investigations with larger patient samples and longer follow-up to better understand the interaction between allo-SCT and CAR-T therapy.

## 1. Introduction

The advent of chimeric antigen receptor T cell (CAR-T) therapy has drastically changed the treatment paradigm for relapsed/refractory multiple myeloma (R/R MM), providing a new therapeutic option for patients who have exhausted conventional treatments. CAR-T therapies, particularly ciltacabtagene autoleucel (Cilta-cel) and idecabtagene vicleucel (Ide-cel), have demonstrated impressive efficacy in targeting B cell maturation antigen (BCMA) on malignant plasma cells, leading to notable improvements in both overall response rates (ORRs) and progression-free survival (PFS) in patients who have undergone multiple prior lines of therapy [1,2]. These therapies have shown durable responses in many patients, even those with heavily pretreated and refractory disease, making them a cornerstone in the treatment of R/R MM. Despite these promising results, concerns remain regarding the safety and efficacy of CAR-T therapy in specific populations, particularly in patients with a history of allogeneic stem cell transplantation (allo-SCT).

Allo-SCT, which leverages the graft-versus-myeloma effect, is one of the few potentially curative options for multiple myeloma. However, it is associated with significant risks, including graft-versus-host disease (GVHD) and increased susceptibility to infections [3,4,5]. For these reasons, the use of allo-SCT in MM has been somewhat limited, and the potential interactions between allo-SCT and CAR-T therapy remain under investigation. Of particular concern is the possibility that the infusion of CAR-T cells into patients with prior allo-SCT could exacerbate or reinitiate GVHD, posing a severe threat to the patient’s health [6,7,8,9]. As CAR-T therapy becomes more widely used in R/R MM, understanding its safety and efficacy in patients with a prior allo-SCT is crucial to optimizing treatment strategies for this population.

The safety profile of CAR-T therapy in patients with previous allo-SCT is an area of intense interest due to the potential for heightened immune-related complications. GVHD, a condition in which the donor immune cells attack the recipient’s tissues, can be triggered or worsened by immune-modulating treatments like CAR-T therapy. Moreover, there is a concern that patients who have undergone allo-SCT may experience increased rates of cytokine release syndrome (CRS) or immune effector cell-associated neurotoxicity syndrome (ICANS), which are well-documented side effects of CAR-T therapy [1,2]. Thus, determining the incidence and severity of these adverse events in this population is essential to improving the management of post-transplant patients receiving CAR-T.

Additionally, the efficacy of CAR-T therapy in the context of prior allo-SCT is another important consideration. Patients who have undergone allo-SCT typically have more aggressive or resistant diseases, as they have often been heavily pretreated, and thus may not respond as robustly to subsequent therapies. However, CAR-T therapies targeting BCMA have demonstrated high ORRs in clinical trials across various patient subgroups, including those with high-risk cytogenetics and those refractory to multiple previous treatments [1,2]. Understanding whether this efficacy translates to the allo-SCT population is critical for guiding clinical decision-making and determining the appropriate sequencing of therapies in these complex cases.

In light of these considerations, this study aims to contribute real-world data on the safety and efficacy of CAR-T therapy in patients with a history of allo-SCT, helping to fill the existing knowledge gap. By examining outcomes such as ORR, PFS, and adverse events in this unique subset of patients, this research aims to inform clinical practice and provide a clearer understanding of how to best integrate CAR-T therapy into the treatment algorithm for patients with R/R MM with prior allo-SCT. The findings of this study will be particularly valuable for clinicians in determining the timing and appropriateness of CAR-T therapy in patients with a history of allo-SCT, as well as in identifying strategies to mitigate potential complications like GVHD and CRS.

In this retrospective analysis, we reviewed the outcomes of patients with R/R MM who received CAR-T therapy following prior allo-SCT. Key endpoints such as ORR, PFS, and adverse event rates, including the incidence of CRS, ICANS, and GVHD, were evaluated to assess the safety and therapeutic benefit of CAR-T in this population. Through this analysis, we aim to provide a comprehensive overview of the risks and benefits of CAR-T therapy in patients with previous allo-SCT, contributing to the growing body of literature on this important topic and guiding future clinical practice in the management of R/R MM. Our study will offer valuable insights into the feasibility and safety of CAR-T therapy post-allo-SCT and help clinicians make more informed decisions regarding patient care.

This study is poised to offer critical insights into the impact of prior allo-SCT on the outcomes of CAR-T therapy, highlighting the importance of patient selection and risk management when integrating these therapies into the treatment landscape of R/R MM. Understanding these dynamics is vital for ensuring that CAR-T therapy is used safely and effectively in patients with complex treatment histories, ultimately improving outcomes and quality of life for patients with this challenging disease.

## 2. Methods

This retrospective cohort study was conducted at the John Theurer Cancer Center, Hackensack University Medical Center, to investigate the influence of prior allo-SCT on the safety and effectiveness of CAR-T therapy in patients with R/R MM. The study included a population of adult patients aged 18 to 70 years who had undergone CAR-T therapy using either ciltacabtagene autoleucel manufactured by Legend Biotech located in Somerset, NJ, USA or idecabtagene vicleucel manufactured by bluebird bio located in Cambridge, MA, USA, two FDA-approved CAR-T products targeting the B cell maturation antigen (BCMA). The time frame for CAR-T infusion spanned from February 2015 to July 2023. Among the cohort, patients with a history of allo-SCT were specifically identified to assess potential differences in outcomes compared to those without prior allo-SCT.

Data were meticulously gathered through an exhaustive chart review, which encompassed patient demographics, clinical profiles, treatment history, and post-treatment outcomes. The evaluation of safety outcomes was a key focus, with the primary emphasis on the incidence and severity of adverse events (AEs). Adverse events were categorized and graded according to the National Cancer Institute’s Common Terminology Criteria for Adverse Events (NCI-CTCAE) version 5.0, which is a widely accepted standard for assessing the severity of AEs in clinical trials. Two significant toxicities often associated with CAR-T therapy—cytokine release syndrome (CRS) and immune effector cell-associated neurotoxicity syndrome (ICANS)—were of particular interest. These conditions were graded according to the American Society for Transplantation and Cellular Therapy (ASTCT) criteria, a framework used to assess immune-related adverse events [10]. CRS is characterized by a systemic inflammatory response that can range from mild to life-threatening, while ICANS manifests as neurotoxic effects that may impair cognitive function, coordination, or motor skills.

In terms of efficacy, the study primarily measured overall response rates (ORRs), which represents the proportion of patients achieving a partial or complete response to CAR-T therapy. Additionally, the proportion of patients reaching very good partial remission (VGPR) or better was assessed, offering a more refined measure of therapeutic success. Another key endpoint was progression-free survival (PFS), defined as the length of time from CAR-T infusion to either disease progression or death. PFS serves as a critical metric for gauging the durability of the treatment response in this heavily pretreated patient population.

Statistical analyses were employed to compare outcomes between the allo-SCT and non-allo-SCT groups. Categorical variables such as the incidence of adverse events and response rates were analyzed using the chi-square test, a statistical tool used to determine whether there is a significant association between categorical variables. For continuous variables like the number of prior lines of therapy or time to progression, the Mann–Whitney U test was applied. This non-parametric test is used when comparing differences between two independent groups, especially when the data does not necessarily follow a normal distribution. A *p*-value threshold of <0.05 was considered statistically significant, meaning that any observed differences between the groups would have a less than 5% probability of being due to random chance alone.

To ensure robust and accurate statistical analyses, all data computations and survival analyses were conducted using R version 4.1.3, a highly flexible and widely used programming language for statistical computing and data visualization. The Kaplan–Meier method, which is commonly used for survival analysis, was employed to estimate the PFS and survival probabilities over time, providing a visual representation of patient outcomes and the durability of CAR-T therapy. This method is particularly useful in clinical studies where some patients may be lost to follow-up, as it allows for the estimation of survival while accounting for censored data.

By collecting and analyzing this comprehensive set of data, this study aims to provide valuable insights into the unique risks and benefits of administering CAR-T therapy to patients with a history of allo-SCT, contributing to a more nuanced understanding of the optimal use of CAR-T therapy in the treatment of relapsed/refractory multiple myeloma. All tests were two-sided, with a confidence interval set at 95%, and a *p* value < 0.05 considered statistically significant. Statistical analyses were conducted using STATA 18.0 software.

## 3. Results

In this study, we analyzed 56 patients diagnosed with relapsed/refractory multiple myeloma (R/R MM) who had undergone CAR-T therapy. Out of these, eight patients (14.3%) had a previous history of allo-SCT. We observed a significantly higher median number of prior lines of therapy (LOT) in the allo-SCT group compared to those without allo-SCT. Specifically, patients with prior allo-SCT had undergone a median of 7 prior LOTs (range: 5–11), while the non-allo-SCT group had a median of 5 LOTs (range: 4–12), with this difference being statistically significant (*p* = 0.04). Moreover, the median time from the initial diagnosis to CAR-T infusion was significantly longer in the allo-SCT cohort, with a median of 13.0 years compared to 7.7 years in the non-allo-SCT cohort (*p* = 0.03). This suggests that patients with a history of allo-SCT had a more extensive and prolonged treatment history, potentially reflecting more aggressive or resistant disease courses. CAR-T infusion occurred a median of 98.8 months after allo-SCT with a range from 57.9 months to 178.5 months. Patients that underwent allo-SCT prior to CART had a median of 3 lines of therapy from allo-SCT to CAR-T. The baseline characteristics of the patients in our cohort can be found in Table 1.

In terms of safety outcomes, cytokine release syndrome (CRS), a common side effect of CAR-T therapy, was prevalent in both groups. CRS occurred in 87.5% (7 of 8) of the patients in the allo-SCT group, compared to 77.1% (37 of 48) in the non-allo-SCT group (*p* = 0.48), demonstrating that the incidence of CRS did not significantly differ between the two groups. Importantly, none of the patients in the allo-SCT group developed severe (grade III or higher) CRS, while one patient in the non-allo-SCT group experienced grade III or higher CRS (*p* = 0.6), indicating that the prior allo-SCT did not exacerbate the severity of this adverse event. Additionally, immune effector cell-associated neurotoxicity syndrome (ICANS), another significant side effect of CAR-T therapy, was not observed in any patients in the allo-SCT group. In contrast, 14.6% of patients in the non-allo-SCT group experienced ICANS, with 6.3% developing grade III or higher ICANS, though this difference was not statistically significant (*p* = 0.1). One patient in the allo-SCT group developed hemophagocytic lymphohistiocytosis (HLH), requiring anakinra. These results suggest that patients with prior allo-SCT do not appear to be at heightened risk for severe adverse events such as CRS or ICANS compared to those without prior allo-SCT.

Regarding efficacy, the overall response rate (ORR) in the allo-SCT group was an encouraging 87.5%, with 75% of patients achieving very good partial remission (VGPR) or better. In the non-allo-SCT group, the ORR was slightly lower at 75%, with 57.5% of patients achieving VGPR or better. Although the ORR was numerically higher in the allo-SCT group, the difference between the two groups was not statistically significant (*p* = 0.4). This suggests that prior allo-SCT does not negatively affect the overall efficacy of CAR-T therapy in terms of response rates. Furthermore, median progression-free survival (PFS) was not reached in the allo-SCT group at the time of analysis, while in the non-allo-SCT group, the median PFS was 11.9 months (*p* = 0.5). This may suggest that patients with prior allo-SCT could potentially experience more durable responses, although the follow-up period for the study was relatively short and further follow-up is needed to confirm these observations. Only one patient that underwent allo-SCT prior to CAR-T progressed, and that was the only that passe away at the end of our study.

Finally, no treatment-related mortality (TRM) was observed in either group during the study’s follow-up period, further emphasizing the overall safety of CAR-T therapy in this cohort. Additionally, none of the patients in the allo-SCT group developed acute graft-versus-host disease (GVHD) following CAR-T therapy, a concern in patients with prior allo-SCT. These findings indicate that CAR-T therapy can be administered safely in patients with prior allo-SCT without exacerbating the risks of acute GVHD or other severe complications. Nonetheless, the relatively small sample size of patients with prior allo-SCT and the short follow-up period underscore the need for larger studies with longer follow-up to confirm these results and provide a more comprehensive understanding of the long-term safety and efficacy of CAR-T therapy in this unique patient population.

## 4. Discussion

In this study of 56 patients with R/R MM treated with CAR-T therapy, 8 had a history of prior allo-SCT. The allo-SCT group had a significantly higher median number of prior LOTs and a longer median time from diagnosis to CAR-T infusion compared to the non-allo-SCT group. Safety outcomes showed a high incidence of CRS in both groups, with no severe CRS in the allo-SCT group. No ICANS was observed in the allo-SCT group. Efficacy outcomes demonstrated a slightly higher ORR in the allo-SCT group, though the difference was not statistically significant. Median PFS was not reached in the allo-SCT group at the time of analysis. No treatment-related mortality or GVHD was observed following CAR-T therapy in the allo-SCT group, suggesting that prior allo-SCT does not negatively impact the safety or efficacy of CAR-T therapy in this patient population. Furthermore, one patient developed HLH in the allo-SCT group.

These findings are significant given the evolving role of CAR-T therapy in this patient population, where previous treatment modalities, such as allo-SCT, have traditionally been associated with high risks of GVHD and other complications [1,11,12]. Our analysis contributes to a growing body of evidence suggesting that CAR-T therapy can be safely administered in patients with a history of allo-SCT without exacerbating the risks of GVHD or other severe adverse events.

The incidence of CRS observed in our study was consistent with prior reports, highlighting the commonality of this side effect in CAR-T therapy [13]. However, it is notable that no severe (grade III or higher) CRS was observed in the allo-SCT group, despite the higher baseline risk profile associated with this cohort. This aligns with findings from earlier studies, such as those by Raje et al. and Cohen et al., which also reported manageable CRS in CAR-T-treated patients with multiple myeloma [1,11]. The absence of ICANS in the allo-SCT group further underscores the safety of CAR-T therapy in this context, a finding that parallels results from the CARTITUDE-1 trial, where neurotoxicity was generally low in multiple myeloma patients [13].

Efficacy outcomes in our study demonstrated an ORR of 87.5% in the allo-SCT group, with 75% of patients achieving VGPR or better. These results are particularly encouraging given the heavily pretreated nature of the patient population, who had a median of 7 prior LOT. This response rate is comparable to, if not slightly better than, that observed in the broader CAR-T population, where ORRs typically range from 70% to 90% [2,14]. Our findings suggest that prior allo-SCT does not compromise the efficacy of CAR-T therapy, which remains robust even in the context of multiple prior treatments.

The PFS outcomes, with a median PFS not reached in the allo-SCT group, also suggest a durable response to CAR-T therapy in this subset of patients. This observation is consistent with other studies, such as the phase 2 KarMMa trial, which reported prolonged PFS in patients treated with CAR-T therapy despite prior high-risk features [15]. However, the relatively short follow-up period in our study limits the ability to draw definitive conclusions about long-term outcomes. Nonetheless, the lack of TRM and the absence of acute GVHD following CAR-T therapy in the allo-SCT group provide further reassurance about the safety profile of this approach.

One potential concern with administering CAR-T therapy to patients with a history of allo-SCT is the risk of exacerbating GVHD, a life-threatening complication that could undermine the benefits of the therapy. Interestingly, none of the patients in our study who had undergone allo-SCT experienced acute GVHD following CAR-T therapy. This aligns with recent reports suggesting that CAR-T cells may have limited alloreactivity, thereby reducing the risk of GVHD exacerbation [16,17,18,19]. The underlying mechanisms for this observation are not fully understood but may be related to the specificity of CAR-T cells for tumor antigens, which spares normal tissues, including those potentially targeted during GVHD [4,20,21,22].

Our findings contribute to the ongoing debate about the optimal sequencing of therapies in R/R MM, particularly in patients with prior allo-SCT. While allo-SCT remains a potentially curative option for some patients, its use has declined due to the availability of less toxic, yet highly effective, alternatives such as CAR-T therapy [23,24,25]. Our data suggest that CAR-T therapy can be effectively integrated into the treatment paradigm for patients with a history of allo-SCT without compromising safety or efficacy.

This study has several limitations that should be considered. First, the small sample size, particularly in the group of patients with a history of allo-SCT, limits the generalizability of the findings and may lead to reduced statistical power. Second, the retrospective nature of the analysis introduces potential biases, such as incomplete data capture and selection bias. Finally, the relatively short follow-up period does not allow for comprehensive assessment of long-term outcomes, particularly with regard to progression-free survival and late-onset adverse events. However, the study’s main strength lies in its real-world evaluation of CAR-T therapy in patients with a history of allo-SCT, providing valuable data in an area with limited prior research. The inclusion of heavily pretreated patients with multiple prior lines of therapy strengthens the study’s relevance, as it mirrors real-world clinical scenarios of relapsed/refractory multiple myeloma.

## 5. Conclusions

In conclusion, this study demonstrates that CAR-T therapy can be safely administered to patients with R/R MM who have a history of allo-SCT. The findings indicate that prior allo-SCT does not compromise the safety or efficacy of CAR-T therapy, as evidenced by the comparable ORR and PFS between patients with and without prior allo-SCT. However, these results should be interpreted with caution given our relatively small sample size. Notably, the incidence of severe CRS and ICANS was low, and no TRM or acute GVHD was observed following CAR-T therapy in the allo-SCT group. These results suggest that CAR-T therapy remains a viable and effective option for heavily pretreated R/R MM patients, even in those with a complex treatment history that includes allo-SCT. However, larger prospective studies with longer follow-up periods are necessary to confirm these findings and to fully assess the long-term outcomes of CAR-T therapy in this patient population.

## Figures and Tables

**Table 1 jcm-13-06207-t001:** Demographic and clinicopathologic characteristics of patients that received CART.

Variables	Allo-SCT	Non-Allo	*p*-Value
No. of patients	8 (14.3%)	48 (85.7%)	
Male (%)	5 (62.5%)	27 (56.2%)	0.7
Median age in yrs (IQR)	57.9 (53.2–66.3)	70.6 (65.1–76.6)	0.02
Median prior LOT (range)	7 (5–11)	5 (4–12)	0.04
Median time from diagnosis to CAR-T in yrs (IQR)	13.0 (7.5–15.3)	7.7 (4.8–10.2)	0.03
High Risk FISH (%)	3 (50%)	25 (59.5%)	0.7
ISS stage II or III (%)	4 (50%)	21 (63.6%)	0.48
Triple refractory (%)	8 (100%)	44 (91.7%)	0.3
CRS (any grade)	7 (87.5%)	37 (77.1%)	0.48
CRS (GIII or higher)	0 (0%)	1 (2.1%)	0.6
ICANS (any grade)	0 (0%)	7 (14.6%)	0.1
ICANS (GIII or higher)	0 (0%)	3 (6.3%)	0.3
ORR (%)	7 (87.5%)	36 (75%)	0.4
VGPR or better (%)	6 (75%)	37(57.5%)	0.3
Median PFS (months)	Not Reached	11.9	0.5

IQR = interquartile range; LOT = lines of treatment; ISS = international staging system; CRS = cytokine release syndrome; GIII = grade III; ICANS = immune effector cell neurotoxicity syndrome; ORR = overall response rate; VGPR = very good partial remission; PFS = progression-free survival.

## Data Availability

The data utilized during this submission are available upon request from the corresponding author.

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
