# Peer review of "Impact of Allogeneic Stem Cell Transplant on Safety and Outcomes of Chimeric Antigen Receptor T Cell (CAR-T) Therapy in Patients with Multiple Myeloma (MM)"

_jcm, 2024, doi:10.3390/jcm13206207_

Round 1

Reviewer 1 Report

Comments and Suggestions for Authors

Suggestions:

- The sample size is too small for statistical comparison. Given this small sample size, the results should be more descriptive.

- Time from allo to CART would be relevant to report

- the authors site that GVHD post CART in patients with a prior allo-HSC is a concern. The CART in patients with ALL should be discussed, as these patients often receive an allo prior to CART

- The conclusions are over-reaching. eg. line 269 - this evidence does not support such a definitive statement. This statement needs to be balanced with the limitations of the sample size.

Comments on the Quality of English Language

The English quality was appropriate

Author Response

Thank you for taking the time out of your busy schedule to review our manuscript. 

- The sample size is too small for statistical comparison. Given this small sample size, the results should be more descriptive.

Re: Thank you for this keen observation, A few changes were made to reflect that. 

- Time from allo to CART would be relevant to report

Re: Thank you for this relevant observation. The information can be found in the abstract and it was added to the results section and highlighted in red.  

- the authors site that GVHD post CART in patients with a prior allo-HSC is a concern. The CART in patients with ALL should be discussed, as these patients often receive an allo prior to CART

Re: Thank you for this comment. However, our patient cohort was only for patients with MM, and the Allo was given as part of MM management. None of our patients had concurrent ALL. 

- The conclusions are over-reaching. eg. line 269 - this evidence does not support such a definitive statement. This statement needs to be balanced with the limitations of the sample size.

Re: Thank you for this valid point and a sentence was added to clarify that, highlighted in red. 

Reviewer 2 Report

Comments and Suggestions for Authors

The authors present a retrospective study collecting their experience in the treatment of R/R MM with CAR-T. The interest is accrued by the presence of a small, but not negligeable, group of MM patients undergoing allogeneic SCT before CAR-T treatment.

Unfortunately, the presentation is sometimes confused. At the same time, some key points are neglected. Moreover, there is a diffuse trend toward overstatement.

In the following paragraphs, I am listing general and particular issues, some involving more than one section, deserving the attention of the authors.  

Abstract

- As far as I understand from the methods, the series included patients receiving CAR-T between 2005 and 2013. At line 38 of the introduction “At a median follow-up of 4.8 months, the ORR was 87.5% in the allo-SCT group versus 75% 38 in the non-allo-SCT group (p=0.4)”. It is unlikely that 4.8 months was the median follow-up of the whole series. In its present form, the sentence is misleading.

At line 39 “Median PFS had not been reached for the allo-SCT group at the time of analysis compared to 11.9 months in the non-allo-SCT group (p=0.5)”. The authors should eventually specify follow up data, both in the abstract and in the text.

Introduction

The statement on line 87 overemphasizes the aim of the paper.

Methods and results.

-Both sections are interspersed with short comments deserving a better location elsewhere.

- In spite of the long accrual time, eight MM allo-SCT recipients undergoing CAR-T therapy are not a negligeable series. They likely derive from a larger MM allo-SCT series. In my opinion, the authors should give some detail about the entire series of MM patients undergoing allo-SCT in their institution, over the same period.

- The authors specify the time interval between transplant and CAR-T therapy only in the abstract; they are expected to do the same also in the text.

- I did not find when allo-SCT was performed. This information is of outmost interest, since CAR-T treatment was delivered starting from 2005 and some patients could have undergone SCT notably earlier, in a quite different era in the history of MM treatment.  

-They should only specify whether any line of treatment was administered between allo-SCT and CAR-T.

-The authors do not offer any details about CAR-T administration and management policy of the main complications.

- As a general remark, data on the clinical outcome of the patients are scant. It could be interesting to know e.g. how many patients were alive or dead, in remission or in progression at the end of the follow up period  

Discussion

- The discussion is diffusely affected by overstatement. One more patient with CRS in the SCT group, would made the difference statistically significant; one patient less with CRS or achieving response in the same group, would make the response or the toxicity rate identical in both groups. They never comment that the limited size of the SCT group make comparison difficult. Moreover, they often comment simple trends overemphasizing their findings.

- The authors legitimately comment the lack of ICANS cases and GVHD in the SCT series, At the same time, a short comment could also be devoted to the single case of hemophagocytic syndrome in the SCT group.

-The median time interval between MM diagnosis and CAR-T therapy is 13 years in the SCT group and 7.7 in the non SCT group. Apart from being a significant difference, it implies that the two population are quite different, not only because they underwent or not underwent SCT, but also due to the evolving strategies in MM treatment over the encompassed time span. Moreover, SCT patients are significantly younger, had received more, likely different, therapies, so making the two population barely comparable. In my opinion, this does not lower the interest of the paper, but should be more thoroughly discussed.

- Line 226 “These findings are significant given the evolving role of CAR-T therapy in this patient population”. The sentence is somewhat misleading

- Line 252, the relatively short follow up; again,

Author Response

The authors present a retrospective study collecting their experience in the treatment of R/R MM with CAR-T. The interest is accrued by the presence of a small, but not negligeable, group of MM patients undergoing allogeneic SCT before CAR-T treatment.

Unfortunately, the presentation is sometimes confused. At the same time, some key points are neglected. Moreover, there is a diffuse trend toward overstatement.

In the following paragraphs, I am listing general and particular issues, some involving more than one section, deserving the attention of the authors.  

Re: Thank you for taking the time out of your busy schedule to review our paper. Your valid points were taken into account and changes were made to improve our paper. 

Abstract

- As far as I understand from the methods, the series included patients receiving CAR-T between 2005 and 2013. At line 38 of the introduction “At a median follow-up of 4.8 months, the ORR was 87.5% in the allo-SCT group versus 75% 38 in the non-allo-SCT group (p=0.4)”. It is unlikely that 4.8 months was the median follow-up of the whole series. In its present form, the sentence is misleading.

Re: Thank you for your comment, however the median follow up was based upon the time of CART T infusion which had been approved mainly as the 5th line of therapy up until 2024. Also the years mentionned in our cohort are 2015 and 2023, not 2005 and 2013 as CART was not available back then. 

At line 39 “Median PFS had not been reached for the allo-SCT group at the time of analysis compared to 11.9 months in the non-allo-SCT group (p=0.5)”. The authors should eventually specify follow up data, both in the abstract and in the text.

Re: The authors are not sure they understand the question as the sentence is self explanatory.  

Introduction

The statement on line 87 overemphasizes the aim of the paper.

 Re: The sentence was rephrased and highlighted in red. 

Methods and results.

-Both sections are interspersed with short comments deserving a better location elsewhere.

- In spite of the long accrual time, eight MM allo-SCT recipients undergoing CAR-T therapy are not a negligeable series. They likely derive from a larger MM allo-SCT series. In my opinion, the authors should give some detail about the entire series of MM patients undergoing allo-SCT in their institution, over the same period.

Re: This is a very valid point. However, going over details of patients with MM that underwent allo SCT is beyond the scope of our current study. The main purpose of our study is to evaluate the safety and efficacy of CART in patients that underwent Allo SCT. Nevertheless, future studies addressing this issue will be carried out. 

- The authors specify the time interval between transplant and CAR-T therapy only in the abstract; they are expected to do the same also in the text.

Re: This was added in the results section and highlighted in red. 

- I did not find when allo-SCT was performed. This information is of outmost interest, since CAR-T treatment was delivered starting from 2005 and some patients could have undergone SCT notably earlier, in a quite different era in the history of MM treatment.  

Re: Thank you for your comment, the time from diagnosis to CART has been provided as well as the time from allo to CART. Furthermore, CART was not available in 2005 and nowhere in our paper, such information can be found. 

-They should only specify whether any line of treatment was administered between allo-SCT and CAR-T.

Re: Thank you for this comment, there was a median of 3 lines of therapy prior to CART, this was added to the results section. 

-The authors do not offer any details about CAR-T administration and management policy of the main complications.

Re: Thank you for this comment, complications were defined based on National Cancer Institute Common Terminology Criteria for Adverse Events (NCI-CTCAE) version 5.0 as mentioned in the abstract and the m,anagment of the complications usded the same principles. 

- As a general remark, data on the clinical outcome of the patients are scant. It could be interesting to know e.g. how many patients were alive or dead, in remission or in progression at the end of the follow up period. 

Re: Thank you for this comment. Only one patient that underwent Allo-SCT prior to CAR-T progressed, and that was the only that passe away at the end of our study and this was added to the results section. 

Discussion

- The discussion is diffusely affected by overstatement. One more patient with CRS in the SCT group, would made the difference statistically significant; one patient less with CRS or achieving response in the same group, would make the response or the toxicity rate identical in both groups. They never comment that the limited size of the SCT group make comparison difficult. Moreover, they often comment simple trends overemphasizing their findings.

Re:  Thank you for this observation, a sentence was added to emphasize that the lower sample size is a weakness of our study. 

- The authors legitimately comment the lack of ICANS cases and GVHD in the SCT series, At the same time, a short comment could also be devoted to the single case of hemophagocytic syndrome in the SCT group.

Re: No HLH was observed in our cohort and a sentence was added in the discussion. 

-The median time interval between MM diagnosis and CAR-T therapy is 13 years in the SCT group and 7.7 in the non SCT group. Apart from being a significant difference, it implies that the two population are quite different, not only because they underwent or not underwent SCT, but also due to the evolving strategies in MM treatment over the encompassed time span. Moreover, SCT patients are significantly younger, had received more, likely different, therapies, so making the two population barely comparable. In my opinion, this does not lower the interest of the paper, but should be more thoroughly discussed.

Re: Very well taken point. However, Allo SCT is not a common practice in MM. For MM patients to undergo allo-SCT, means that their disease was very difficult to manage. 

- Line 226 “These findings are significant given the evolving role of CAR-T therapy in this patient population”. The sentence is somewhat misleading

 Re: This was nuanced by the limitations of the study that can be found in the last paragraph of the discussion. 

- Line 252, the relatively short follow up; again,

Re: Very well taken point. This was added in the limitations section. 

Reviewer 3 Report

Comments and Suggestions for Authors

Dear author, after reviewing your article, I am sending you my comments on the matter. I believe that the methodology should be better explained, especially the instruments used, indicating whether they are supported or not and what they allow to be measured or evaluated. In the discussion section, I believe that it is necessary to include the experiences of other researchers and to engage in a true academic-scientific discussion that supports and contrasts their results.

That is all I can tell you.

Kind regards.

Author Response

Dear author, after reviewing your article, I am sending you my comments on the matter. 

Re: Thank you for taking the time out of your busy schedule to review our manuscript. 

I believe that the methodology should be better explained, especially the instruments used, indicating whether they are supported or not and what they allow to be measured or evaluated. 

Re: Thank you for this valid comment. The following sentence was adde to the methods section: All tests were two-sided, with a confidence interval set at 95%, and a P value < 0.05 considered statistically significant. Statistical analyses were conducted using STATA 18.0 software.

In the discussion section, I believe that it is necessary to include the experiences of other researchers and to engage in a true academic-scientific discussion that supports and contrasts their results.

Re: Thank you for this keen observation.However, the lack of previous data on this matter made difficult for us to compare our results and this would be considered as a pilot study. 

That is all I can tell you.

Kind regards.

Re: Thank you again!

Round 2

Reviewer 1 Report

Comments and Suggestions for Authors

The authors have sufficiently addressed the comments. There are however some typos in the additional text. I would recommend a range or SD with the median time from allo to CART.

Comments on the Quality of English Language

There are some typos in the additional text.

Author Response

The authors have sufficiently addressed the comments. There are however some typos in the additional text.

Re: Thank you very much for your comment. 

I would recommend a range or SD with the median time from allo to CART.

Re: Thank you for this comment and the range was added. 

Reviewer 2 Report

Comments and Suggestions for Authors

I apologize to the authors and to the editor for repeatedly writing 2005-2013 instead of 2015-2023. Nevertheless, my remarks were intended to the 2015-2023 period, i.e. starting about ten years ago.

The authors did their best to meet many points. Moreover, I understand that some data could not be retrieved. Besides, the authors legitimately disagree on some of my comments. Nevertheless, minor questions remain still open.

- The paper is presented as a real-life report. In my opinion, some more clinical detail could have been offered. The authors legitimately disagree. I do not assume to be right.

- The Methods and Results sections are interspersed with short comments. In my opinion, the authors could try to find out whether they are necessary. Likely, they disagree but they did not answer the question. Again, I do not assume to be right.

- The last paragraph in the Results section should be moved to the Discussion.

- Abstract Line 36, “CRS occurred in 85.7% of the allo-SCT group”. Results line 184, “CRS occurred in 87.5% (7 of 8)”. 87.5 is more likely to be correct.

- Abstract Line 36 “One patient in the allo-SCT group developed hemophagocytic lymphohistiocytosis (HLH) requiring anakinra”. No mention of HLH in the Results. In their reply the authors state “No HLH was observed in our cohort and a sentence was added in the discussion.”. The added sentence is on line 242 “Furthermore, no hemophagocytic lymphohistiocytosis was”. Unfortunately, the sentence is truncated, at least in the version I can read. Which is the truth?

Author Response

I apologize to the authors and to the editor for repeatedly writing 2005-2013 instead of 2015-2023. Nevertheless, my remarks were intended to the 2015-2023 period, i.e. starting about ten years ago.

The authors did their best to meet many points. Moreover, I understand that some data could not be retrieved. Besides, the authors legitimately disagree on some of my comments.

Re: Thank you very much for recognizing the relevance of our study and acknowledging our responses. 

 Nevertheless, minor questions remain still open.

- The paper is presented as a real-life report. In my opinion, some more clinical detail could have been offered. The authors legitimately disagree. I do not assume to be right.

Re: Thank you for this observation and agreeing with us. 

- The Methods and Results sections are interspersed with short comments. In my opinion, the authors could try to find out whether they are necessary. Likely, they disagree but they did not answer the question. Again, I do not assume to be right.

Re: Thank you for this comment. However, the methods are clearly stated and the results are clearly stated as well. 

- The last paragraph in the Results section should be moved to the Discussion.

Re: The authors totally agree with the reviewer and this change was made. 

- Abstract Line 36, “CRS occurred in 85.7% of the allo-SCT group”. Results line 184, “CRS occurred in 87.5% (7 of 8)”. 87.5 is more likely to be correct.

Re: The authors totally agree with the reviewer and this change was made. 

- Abstract Line 36 “One patient in the allo-SCT group developed hemophagocytic lymphohistiocytosis (HLH) requiring anakinra”. No mention of HLH in the Results. In their reply the authors state “No HLH was observed in our cohort and a sentence was added in the discussion.”. The added sentence is on line 242 

“Furthermore, no hemophagocytic lymphohistiocytosis was”. Unfortunately, the sentence is truncated, at least in the version I can read. Which is the truth?

Re: The authors totally agree with the reviewer, this was a typo and the change was made. One patient had HLD requiring ankinra. This was added in the Results as well.